# Comparison of Skeletal and Dental Changes Obtained from a Tooth-Borne Maxillary Expansion Appliance Compared to the Damon System Assessed through a Digital Volumetric Imaging: A Randomized Clinical Trial

**DOI:** 10.3390/jcm9103167

**Published:** 2020-09-30

**Authors:** Hye Jin Nam, Silvia Gianoni-Capenakas, Paul W. Major, Giseon Heo, Manuel O. Lagravère

**Affiliations:** Faculty of Medicine and Dentistry, Department of Dentistry, Orthodontic Program, University of Alberta, ECHA, 11405—87th Avenue, Edmonton, AB T6G 1C9, Canada; hnam@ualberta.ca (H.J.N.); major@ualberta.ca (P.W.M.); gheo@ualberta.ca (G.H.); manuel@ualberta.ca (M.O.L.)

**Keywords:** orthodontics, palatal expansion technique, cone-beam computed tomography

## Abstract

The purpose of this study was to evaluate and compare dental and skeletal changes associated with the Damon and Rapid Maxillary Expander (RME) expansion using Cone-Beam Computed Tomography (CBCT). Eighty-two patients, from The University of Alberta Orthodontic Clinic, were randomly allocated to either Group A or B. Patients in Group A received orthodontic treatment using the Damon brackets. Patients in Group B received treatment using the Hyrax (a type of RME) appliance. CBCT images were taken two times (baseline and after expansion). The AVIZO software was used to locate 18 landmarks (dental and skeletal) on sagittal, axial, and coronal slices of CBCT images. Comparison between two groups showed that transverse movement of maxillary first molars and premolars was much greater in the Hyrax group. The lateral movements of posterior teeth were associated with buccal tipping of crowns. No clinically significant difference in the vertical or anteroposterior direction between the two groups was noted. Alveolar bone next to root apex of maxillary first premolar and molar teeth showed clinically significant lateral movement in the Hyrax group only. The comparison between two groups showed significantly greater transverse expansion of the first molar and first premolars with buccal tipping in the RME group.

## 1. Introduction

A deficiency in maxillary transverse width has considerable implications in orthodontic treatment. Having an inadequate transverse relationship between a maxilla and a mandible can lead to increased cervical wear (abfraction), dental arch crowding, and possible negative effects on patients’ smile esthetics and airways [1,2]. Rapid Maxillary Expansion (RME) is one method to treat narrow maxillary arches by separation of the midpalatal suture. Hyrax is one of the most commonly used RME appliances, which has a screw device positioned across a palatal vault, and the device is activated by turning the screw until the desired maxillary expansion is achieved [3]. Studies have shown side effects associated with the appliance such as active root resorption on the buccal side of the first premolars used for anchorage [4,5]. The device also has been reported to expand the upper arch mainly by tipping and extruding the maxillary posterior teeth [6,7,8]. 

The Damon appliance (Ormco, Orange, CA, USA) is one of the self-ligating bracket systems that claims to have benefits over traditional bracket systems. The system claims to apply just enough force to generate an “optimal force zone” so that dental arch expansion can be achieved without the use of a mechanical expander [9]. The Damon philosophy argues that the light force produced by the system allows the connective tissue and alveolar bone to follow tooth movement that results in a more predictable expansion of the maxillary arch in non-extraction cases, through the concept of stimulating cellular activity without damaging the vascular net of the periodontium [10]. As a result, common side effects seen in the RME such as dental tipping, extrusion, and root resorption may be minimized with the Damon System [11]. In addition to the benefits listed above, the company also advertises the “lip bumper” effect on the incisors (minimizing anterior tipping), faster treatment, greater comfort, and better facial esthetic results [9,10]. 

Despite numerous claims of clinical advantages made by the Damon bracket manufacturer, the evidence behind their philosophy is weak [10]. Few studies evaluated the intermolar and intercanine width changes using Damon device; however, only one [12] study evaluated the skeletal and dental changes between Damon and traditional RME treatment. Yu et al. concluded that both treatment methods could increase the arch width along with buccal tipping of posterior teeth [12]. Their study had a small sample size (19 patients total), and the analysis was based on 2-dimensional imaging, which often does not provide accurate measurement of craniofacial changes. With the advance of 3-dimensional (3D) imaging techniques and increasing availability in many offices, the need for 3D analysis of skeletal and dental changes between two treatment methods is needed. Thus, the objective of this study was to compare dental and skeletal changes obtained from Damon and RME expansion, using Hyrax, through 3D imaging. Our null hypothesis is that there is more skeletal expansion in the use of RME than with Damon, despite the Damon System’s claims that the buccal corridor can be widened.

## 2. Experimental Section

This prospective 2-arm parallel group, randomized clinical trial was done at the Orthodontic Clinic in the University of Alberta (Alberta, Canada) with an allocation ratio of 1:1. The trial was approved by the University of Alberta Research Ethics Board (Pro00013379). In order to compare two different appliances, a minimum sample size of 44 patients per group was calculated to be needed when the effect size index was 0.70, α = 0.05, and power = 0.90 [13].

Inclusion criteria were adolescents from 11 to 16 years of age with diagnosis of maxillary transverse deficiency with unilateral or bilateral crossbite requiring maxillary expansion. Patients were in permanent dentition. All patients had a minimum of 5 mm maxillary constriction determined by calculating the differences between intermolar widths of maxilla and mandible (palatal cusp tips of upper molars to the central fossae of lower molars). Exclusion criteria included patients with syndromic characteristics, systemic diseases, or history of previous maxillary expansion/orthodontic treatment. Each patient had an orthodontic clinical examination and a Cone Beam Computed Tomography (CBCT) (iCAT, Imaging Science International, Hatfield, PA, USA) prior to treatment and 6 months after retention-period. The parameters used for the CBCTs were large field of view 16 × 13.3 cm, voxel size 0.30 mm, 120 kVp, 18.54 mAS, and 8.9 s. A person, external to the research group, generated random number blocks for all patients using an Excel worksheet to randomly allocate patients to each group, once they accepted participation in the study.

Patients in Group A received orthodontic treatment using the Damon System (0.022 inch dimension, self-ligating brackets). Initial alignment was done sequentially with Insignia prefabricated 0.014 NiTi, 0.016 NiTi, and 0.014 × 0.025 NiTi archwires in Damon Arch Form. In addition, as per recommendation by the company’s treatment consultant, patients wore crossbite elastics full-time until maxillary expansion was overcorrected by 20% (buttons were bonded on palatal cusp surface of upper first molar and first premolar, elastics used were 3/16 inch, 2-ounce force). The 20% overcorrection aimed to expand the maxillary arch until palatal cusps of upper molars were at the level of the buccal cusps of lower molars. The amount of expansion acquired was determined by measuring the interpalatal cusp distance of the upper first molars and the interfossa distance of the lower first molars. The difference would be the expansion needed and 20% extra was added. At the time overcorrection was achieved, the patients were instructed to wear the elastics at night for 6 months. Bite ramps were also placed on palatal cusps of upper first molars.

Group B treatment consisted of maxillary expansion using the Hyrax appliance attached to the upper first premolars and first permanent molars. This appliance should be activated with one turn of the screw/twice a day (0.25 mm per turn, 0.5 mm daily) until 20% over correction was achieved. On the same day of the Hyrax insertion, non-self-ligating brackets (Insignia, Mini Diamond, Ormco, Orange, CA, USA) were bonded from maxillary right canine to left canine and mandibular right first molar to left first molar. The bonding was done following the Insignia (Ormco, Orange, CA, USA) protocol of indirect bonding set-up and placement. After completion of the active expansion treatment, the screw was tied with a ligature, and the Hyrax stayed passive for a six-month retention period.

The recruitment began in May 2010 and ended in May 2016. Initially, 90 patients had pretreatment records. Four patients in each group decided to not start treatment due to financial concerns. However, once the treatment started, there were no dropouts. A total of 41 patients in the Damon group and 41 patients in the Hyrax group started and completed treatment. Table 1 shows the demographic characteristics of patients who completed the study. The mean age for the Damon group was 13.5 years and for Hyrax group was 13.3 years. The Damon group contained 13 male and 28 female patients, whereas the Hyrax group had 19 males and 22 female patients (Table 1). All diagnostic records were coded. The principal investigator was blinded with respect to treatment group and timing of each record when analyzing the diagnostic records.

Raw CBCT DICOM images were loaded to the Avizo software 8.0 (Visualization Sciences Group, Burlington, MA, USA), which was used with the ISO-surface and exposure of 300–1000 as a setting. All landmarks were identified using a 0.5 mm diameter spherical marker. A total of 25 dental and skeletal 3D landmarks were identified on CBCT images, and out of 25 landmarks, 7 landmarks (Foramen magnum, Right/Left Foramen Ovale, Right/Left Foramen Spinosum, Right/Left External Auditory Meatus) were only used as 3D anatomical references for superimposition. Eighteen landmarks used for the analysis are listed and defined with their acronyms (Table 2 and Table 3). Each landmark was located using sagittal, axial, and coronal multiplanar slices with x, y, and z coordinates. Once landmarks for all the coordinates were obtained in AVIZO software, the collected data were exported as an Excel spreadsheet. The Matlab R2018b (Matrix Laboratory, Natik, MA, USA) software then used the Excel data to create new coordinates and generate optimization. The cranial base landmarks were used to align the Excel data into three planes based on a 3D Cartesian coordinate system. A landmark-derived superimposition technique was used to superimpose all traced CBCT images, a few groups had previously published the detailed description and error associated with this superimposition technique [14,15]. As a result of superimposition and optimization, a positive change in X, Y, and Z coordinates represent left, front, and upward directions respectively. A negative change in X, Y, and Z coordinates represent right, back, and downward directions respectively.

### Statistical Analysis

All statistical analyses were performed using the SPSS analysis software (Version 23 IBM, Armonk, NY, USA). Ten CBCT files were randomly chosen from 164 patients’ CBCT files (82 patients × 2 files for initial/final) for the reliability test. All 25 landmarks (including cranial base landmarks used for superimposition) were traced 3 times over 2-week intervals in between trials. Intraclass Correlation Coefficient (ICC) for all landmarks were 1.00 with the 95% confidence interval of [1.00, 1.00] except for the right external auditory meatus with ICC of 0.99 [0.96, 1.00] and left external auditory meatus with ICC of 0.98 [0.95, 1.00].

The Repeated Measures Mixed ANOVA was performed to assess within group and between group differences from T1 to T2 (Table A1 and Table A2). In order to investigate the specific interaction between the coordinate, landmark, treatment time, and treatment group, a posthoc test was performed. The Bonferroni test was applied to decrease the risk of alpha-error accumulation due to multiple testing.

## 3. Results

Each landmark was analyzed in three different planes, with a total of 54 landmark/coordinate combinations. One mm was chosen to be the threshold for clinical significance because all landmarks used for the analysis except for the right greater palatine foramen had an intra-rater measurement error of <1 mm. Thus, the measurement error cannot be larger than the means between groups to become clinically significant. Previous studies also noted that the intraexaminer’s mean difference for CBCT landmarks was less than 1.50 mm [16]. Furthermore, many clinicians would agree that a less than 1 mm change does not have a meaningful therapeutic effect in orthodontics. Such a small change would not affect the majority of clinicians’ diagnosis and treatment planning decisions.

Table 4 summarizes landmarks that showed statistically significant differences between the two treatment groups. In terms of skeletal landmarks, the alveolar bone next to right and left maxillary first premolars and the molar root apex showed more lateral movement in the Hyrax group. The left and right greater palatine foramen also showed more lateral movement in the Hyrax group although this difference was not clinically significant. Dental landmarks representing upper right and left first molars and first premolars showed the largest difference between treatment groups. The pulp chamber of the upper right first molar (UR6P) in the Hyrax group moved 2.36 ± 0.64 mm more right than in the Damon group while the pulp chamber of the upper left first molar (UL6P) moved 2.89 ± 0.66 mm more left than in the Damon group. The pulp chamber of the upper right first premolar (UR4P) in the Hyrax group moved 1.51 ± 0.75 mm more right than in the Damon group while the pulp chamber of upper left first premolar (UL4P) moved 2.35 ± 0.82 mm more left than in the Damon group. The only landmarks with significant differences between groups in the anteroposterior direction were the upper right first molar root apex (UR6R) and upper right first molar alveolus (UR6A); however, these differences were not clinically significant. There were no statistically significant differences between groups in the Z coordinate.

For the within-group analysis, regardless of the treatment group, on average, all landmarks showed the greatest mean change in the Z coordinate (vertical dimension) (Table 4). In the Damon group, the only clinically significant mean change in the X coordinate was UR4P (−1.15 ± 0.53 mm). In the Hyrax group, landmarks representing upper right and left first molars and premolars showed the largest transverse change over treatment. In both groups, the pulp chamber of the mandibular right and left first molars showed the greatest mean change in the Y coordinate (Table A1).

## 4. Discussion

Comparison between these two treatment groups revealed that most clinically significant changes occurred in the dental landmarks. Skeletal landmarks such as right and left greater palatine foramen showed more lateral movement in the Hyrax group but the values were not clinically significant. This finding agrees with previous studies that treatment effects from the RME device are primarily on the dental structures compared to the skeletal structures [17,18].

The Hyrax group showed more transverse changes than the Damon group. On average, there was 4–5 mm more transverse expansion in the Hyrax group. The Hyrax device is cemented in the patient’s mouth anchored to upper premolar and molar teeth bilaterally. Therefore, the largest mean difference was found in upper right and left first premolar and molar regions (Table 4). Indeed, the pulp chamber of upper right and left molars showed more lateral movements compared to their root apex (Table A1). This finding is in agreement with previous studies that buccal tipping of molars occurs during expansion instead of during bodily movement of tooth [17]. Since both the Hyrax and Damon systems are used to widen maxillary arches transversely, the greatest difference between the two groups was noted in the X-coordinate. In terms of vertical changes, it should be considered that extrusion of upper molars would cause clockwise rotation of the mandible, retruding it, and intrusion of the upper molars would have the opposite effect. However, no clinically significant changes were observed in vertical (Z-coordinate) and anteroposterior (Y-coordinate) directions.

Furthermore, the alveolar bone next to the root apex of maxillary first premolar and molar showed clinically significant lateral movement in the Hyrax group only (Table A1). However, the magnitude of alveolar bone’s lateral movement was smaller than the changes in pulp chamber location. For example, maxillary right first molar pulp chamber moved 2.97 ± 0.46 mm laterally while its alveolar bone next to root apex moved 0.94 ± 0.60 mm (Table A1). This may suggest that lateral movement of maxillary posterior teeth could produce buccal bone apposition; however, the amount of bone apposition is far less than the amount of tooth movement, and there is net thinning of the buccal alveolar bone [19,20]. Furthermore, the right greater palatine foramen moved laterally by 0.78 ± 0.28 mm, which is roughly the same amount of movement as the alveolar bone. Thus, one could also argue that 0.94 mm lateral movement of the alveolar bone next to first molar can be merely from the overall buccal movement of the bony segment from the expansion at the suture line.

Caution must be taken when analyzing the within-group analysis. Since there is no control group, the changes over six months cannot be distinguished between the growth of patients and treatment effects. In both treatment groups, the largest change was in the Z coordinate (downward vertical displacement of all landmarks) between the initial and 6-month records. Previous research has shown that with maxillary expansion, anchoring maxillary teeth follow downward maxillary displacement, and as a consequence, vertical downward and backward rotation of the mandible is observed [21]. However, there was a large variation in the magnitude of vertical change among individuals. Since the study was based on adolescent patients, vertical change is most likely related to growth and not from maxillary expansion.

In the Damon group, bite ramps were placed on upper first molars, and patients wore full-time crossbite elastics (from upper first molars and premolars to lower first molars) to help with arch expansion based on a recommendation from the Damon System’s expert. Therefore, lower first right and left molars were included in this analysis to assess the effect of crossbite elastics and maxillary expansion on lower dentition. There was clinically significant downward and forward movement of the lower first molar’s pulp chamber in both treatment groups but no clinically significant changes in transverse dimension. The vertical and anteroposterior movement of lower molar teeth are likely mainly due to the patient’s growth. It seems that there is no clear effect of crossbite elastics and maxillary expansion on lower molars in the transverse dimension.

Furthermore, through the bite elastics, the Damon archwires are designed to produce forces for wider arch development [9]. However, most transverse changes were noted only in the Hyrax group despite the Damon System’s claim that their device can result in predictable maxillary dental expansion. The Damon System also argues for its capability of reducing negative side effects seen in Hyrax appliances such as downward displacement of maxillary teeth [11]. However, similar vertical displacement of maxillary teeth was noted in both Damon and Hyrax groups in this study although the clear distinction between growth and treatment effects is not clear in the study.

### Limitations

CBCT has limitations for analyzing alveolar bone thickness. The voxel size (spatial resolution) and partial volume effect of CBCT can influence the accuracy of the measurement [22,23]. Since the voxel size used for the study was 0.3 mm, unless the buccal bone is at least 0.6 mm thick, the alveolar bone is not discernable [19]. Identification of alveolar bone defects such as fenestration is three times more likely to be detected in CBCT compared to direct skull analysis, suggesting false positive change based on CBCT [22]. If the voxel lies between two objects of different densities such as alveolar bone and tooth apex, the image will reflect the average density value, and therefore, precise tooth apex identification can be more challenging.

In the Damon group, some clinicians could argue that 0.014 × 0.025 NiTi archwire is not strong enough to generate force for arch expansion. However, if Damon’s claim that light and continuous force can translate teeth, the 0.014 × 0.025 NiTi archwire should generate enough force for adequate arch expansion. Yet, this claim was not supported by our findings. In addition, the time records were taken may not be long enough for the formation of alveolar bone in response to tooth movement [24]. A long-term post retention study is indicated to assess the possible bone formation after treatment is completed.

## 5. Conclusions

This study showed that more transverse expansion was noted in the Hyrax group than in Damon group. No clinically significant changes were observed in vertical (Z-coordinate) and anteroposterior (Y-coordinate) directions. The largest transverse changes were observed in maxillary molar and premolar teeth with buccal tipping movement. The alveolar bone next to the root apex of maxillary first premolar and molar showed clinically significant lateral movement in the Hyrax group suggesting a possible bone apposition following teeth movement.

## Figures and Tables

**Table 1 jcm-09-03167-t001:** Demographic characteristics of the sample.

Variable	Group A (Damon) Mean (SD) or *n* (%)	Group B (Hyrax) Mean (SD) of *n* (%)	*p*-Value
Age (y)	13.8 (1.6)	13.3 (1.5)	0.188
Sex			0.174
Male	13 (31.7%)	19 (46.3%)	
Female	28 (68.3%)	22 (53.7%)	

**Table 2 jcm-09-03167-t002:** Dental and skeletal landmarks used in the study.

Acronym	Landmark Full Name	Acronym	Landmark Full Name
UR6P	Upper Right First Molar Pulp Chamber	UL6R	Upper Left First Molar Mesial Root Apex
UR6R	Upper Right First Molar Mesial Root Apex	UL6A	Upper Left First Molar Alveolar Bone
UR6A	Upper Right First Molar Alveolar Bone	LL6P	Lower Left First Molar Pulp Chamber
LR6P	Lower Right First Molar Pulp Chamber	LL6R	Lower Left First Molar Mesial Root Apex
LR6R	Lower Right First Molar Mesial Root Apex	UL4P	Upper Left First Premolar Pulp Chamber
UR4P	Upper Right First Premolar Pulp Chamber	UL4R	Upper Left First Premolar Root Apex
UR4R	Upper Right First Premolar Root Apex	UL4A	Upper Left First Premolar Alveolar Bone
UR4A	Upper Right First Premolar Alveolar Bone	UL6P	Upper Left First Molar Pulp Chamber
RGP	Right Greater Palatine Foramen	LGP	Left Greater Palatine Foramen

**Table 3 jcm-09-03167-t003:** Craniofacial landmarks identification.

Landmark Description	3D View (Upper Left), Axial View (Upper Right), Sagittal View (Lower Left), Coronal View (Lower Right)
Upper First Molar Pulp Chamber =center of largest cross-sectional pulp chamber area	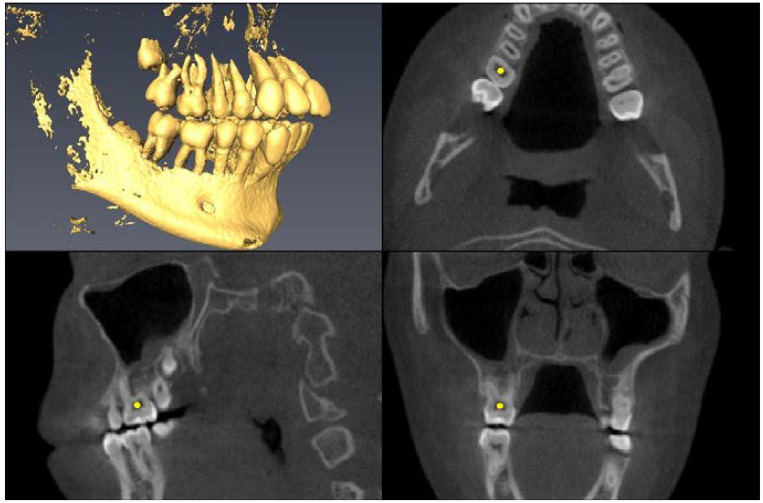
Upper First Molar Mesial Root Apex =apex of mesio-buccal root	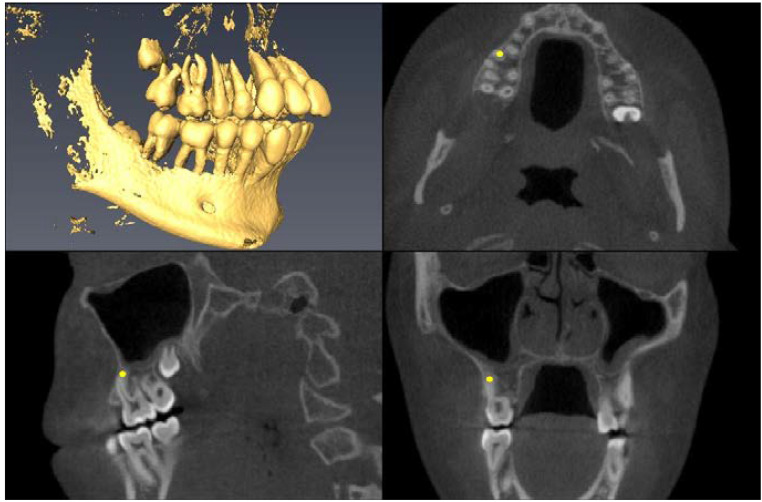
Upper First Molar Alveolar Bone =alveolar bone next to mesial root apex	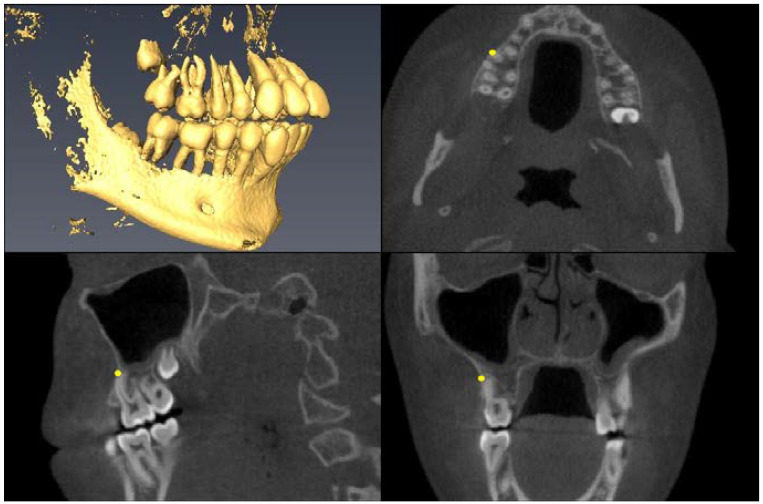
Lower First Molar Pulp Chamber =center of largest cross-sectional pulp chamber area	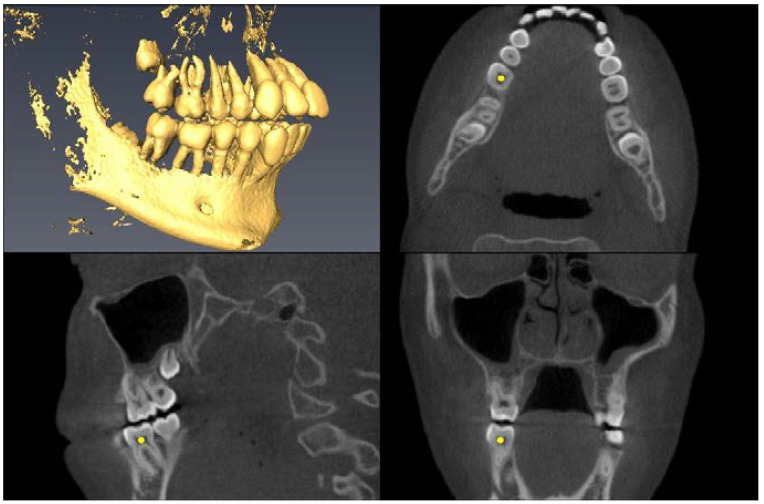
Lower First Molar Mesial Root Apex =apex of mesial root	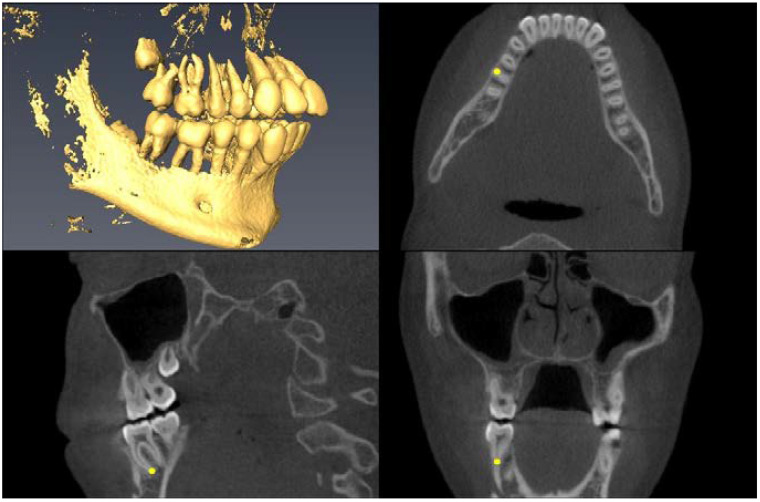
Upper First Premolar Pulp Chamber =center of largest cross-sectional pulp chamber area	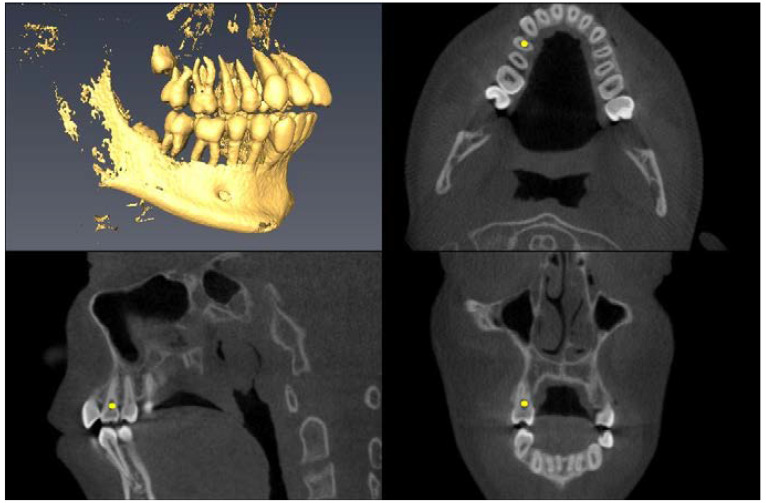
Upper First Premolar Root Apex =apex of mesial root	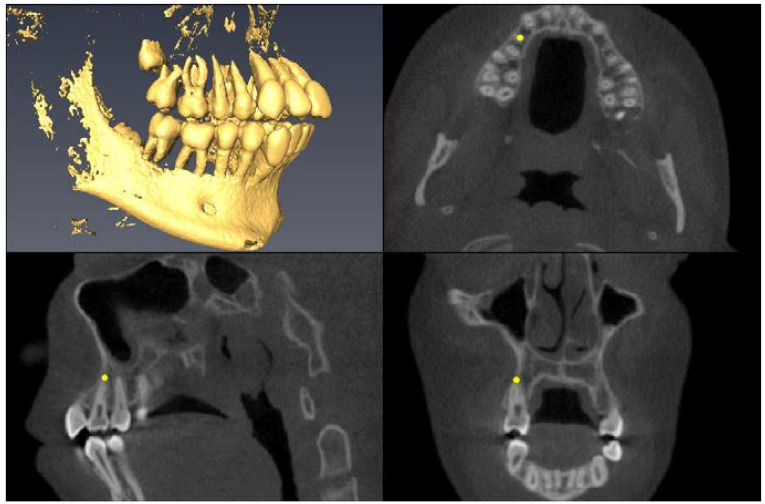
Upper First Premolar Alveolar Bone =alveolar bone next to mesial root apex	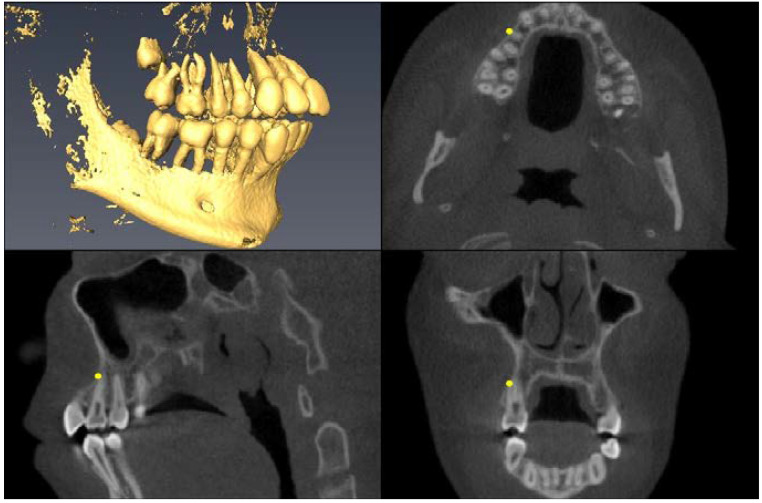
Greater Palatine Foramen =center of largest cross-sectional foramen area	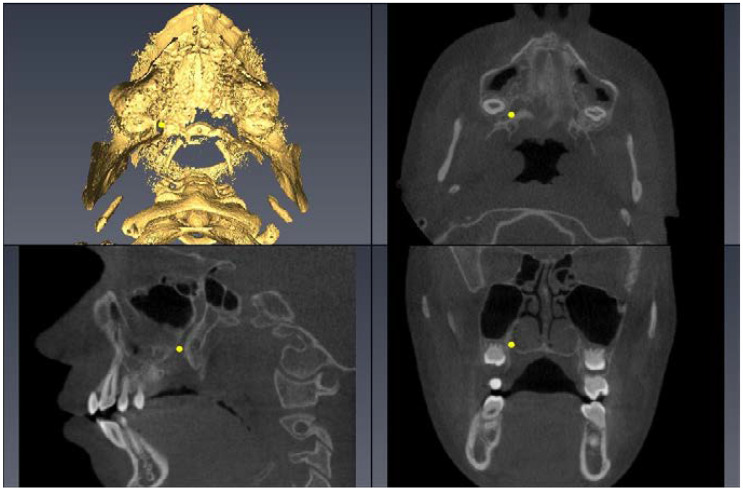

**Table 4 jcm-09-03167-t004:** Summary of statistically significant findings between treatment groups.

Landmark	Mean Difference (Damon-Hyrax)	95% Confidence Interval		Significance	
		Lower Bound	Upper Bound	*p*-Value	Clinical (>1 mm)
Transverse Changes (X coordinate)					
UR6P	2.36	1.71	3.00	<0.0001	Yes
UR6R	2.09	1.38	2.79	<0.0001	Yes
UR6A	1.28	0.42	2.14	0.004	Yes
UR4P	1.51	0.76	2.26	<0.0001	Yes
UR4R	3.01	2.26	3.75	<0.0001	Yes
UR4A	2.00	1.26	2.74	<0.0001	Yes
UL6P	−2.89	−3.55	−2.22	<0.0001	Yes
UL6R	−2.21	−2.89	−1.54	<0.0001	Yes
UL6A	−1.32	−2.00	−0.64	<0.0001	Yes
UL4P	−2.35	−3.17	−1.52	<0.0001	Yes
UL4R	−3.29	−4.10	−2.47	<0.0001	Yes
UL4A	−2.52	−3.34	−1.71	<0.0001	Yes
RGP	0.46	0.06	0.86	0.025	No
LGP	−0.65	−1.05	−0.25	0.002	No
Anteroposterior Changes (Y coordinate)					
UR6R	0.98	0.24	1.73	0.010	No
UR6A	0.99	0.24	1.74	0.011	No
Vertical Changes (Z coordinate)					
None					

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
