# Peer review of "Comparison of Skeletal and Dental Changes Obtained from a Tooth-Borne Maxillary Expansion Appliance Compared to the Damon System Assessed through a Digital Volumetric Imaging: A Randomized Clinical Trial"

_jcm, 2020, doi:10.3390/jcm9103167_

Round 1
Reviewer 1 Report
Dear Authors,
your work is well designed and sounds interesting to the readers.
I notice that you haven't considered the molar relationship and the rotation of the mandible that could occur with this treatment and other vertical changes. Please clarify this aspects considering that a little extrusion of the upper molars causes an antero-rotation of the mandible, improving the Class II and worsening the Class III relationship, and an inhibithion of eruption of the upper molar is responsable of a postero-rotation of the mandible.
Regarding the skeletal effects in the maxilla, you considered the Greater Palatine Foramen as reference point.
Could you support the decision to choose this landmark?
Why you didn't choose the Jugal process as reference point? (Point J is defined at the crossing on the zygomatic process on the maxillary with the outline of the tuberosity).
Line 239: check "special resolution". Do you mean "spatial resolution"?
Could you please improve the conclusions?
Author Response
Dear Reviewer,
Thank you for your comments and inquiries, they always improve the quality of our work. Please see below the responses for each point.
" notice that you haven't considered the molar relationship and the rotation of the mandible that could occur with this treatment and other vertical changes. Please clarify this aspects considering that a little extrusion of the upper molars causes an antero-rotation of the mandible, improving the Class II and worsening the Class III relationship, and an inhibithion of eruption of the upper molar is responsable of a postero-rotation of the mandible."
Thanks for the suggestions, we feel that the reviewer did bring a good point, although the reaction of the mandible would be the opposite as stated, we have added a comment as the one the reviewer mentioned and used a reference to sustain it.
"In terms of vertical changes, it should be considered that extrusion of upper molars would cause clockwise rotation of the mandible, retruding it, and intrusion of the upper molars would have the opposite effect as Kim et al. found in their study."
Regarding the skeletal effects in the maxilla, you considered the Greater Palatine Foramen as reference point. Could you support the decision to choose this landmark?
Why you didn't choose the Jugal process as reference point? (Point J is defined at the crossing on the zygomatic process on the maxillary with the outline of the tuberosity).
Indeed, we chose those landmarks being reliable in locating them in the maxilla and represented a fixed structure in the maxilla that we could identify easily. Point J could have been a good option, unfortunately, this point is for use in 2D imaging (Antero-posterior cephalograms) since it is the overlap of the 2 skeletal structures the reviewer mentions, thus in 3D this is not happening.
Could you please improve the conclusions?"
We have added more detail on the conclusion to reflect better our results
We corrected Line 239: check "special resolution" to "spatial resolution"
Sincerely,
Silvia Gianoni-Capenakas (On behalf of the authors)
Reviewer 2 Report
An extensive review of orthodontic changes comparing maxillary expansion by the Damon System assessed through digital volumetric imaging using cone-beam computed tomography (CBCT) in 82 patients divided into two groups. Group A received Damon brackets treatment. Group B received the Hyrax appliance.
No clinically significant difference in the vertical or antero-posterior direction between the two groups was observed. However, there was significantly greater transverse expansion of the first molar and first premolars with buccal tipping in the Hyrax group.
An excellent review.
Author Response
Dear Reviewer,
Thank you for your comments.
Sincerely,
Silvia Capenakas (on behalf of authors)
Reviewer 3 Report
Dear authors,
thank you for the opportunity to review your manuscript. It was clearly written and thoroughly understandable. Nevertheless, I have some minor suggestions for changes before publication.
line 11:one full stop to much at the end of affiliation
line 50: an "a" is missing in "mechnical"
line 59: one "that" too much
line 72: Why was it a retrospective study? Randomization and allocation to intervention (Damon / Hyrax) was done before intervention, so I guess it was a prospective RCT?
line 73/74: Since Ethics Boards in Europe are very restrictive towards taking CBCTs beyond clinical indication, congratulations for this approval.
line 85: Please specify the exact time point for the CBCT after expansion. Was it after overcorrection was achieved or after the 6 month "retention"-period? Line 214 indicates the latter, but please specify in the Material and Methods section.
line 119, Table 1: Age in group A is not centered within the column.
line 129: Please add the software version number of the Matlab software. Same for line 139, SPSS software.
line 137, Table 2: Please check that acronyms in the third column are also highlighted in bold font. To enhance readability and overview, please check if all right landmarks could be on one side and all left landmarks in the corresponding line on the other side.
line 149: I understand the use of Bonferroni correction to reduce the risk of alpha-error accumulation due to multiple testing. But writing that the research design was done in an unplanned manner sounds real bad because your research design in fact is very well planned :)
line 181: I guess it has to be Table A1?
line 223: Corresponding to line 94, I guess criss-cross elastics were worn on first molars and first premolars?
References: Please check the reference list thoroughly, e.g. reference 2 misses the last page number of the article, some different spellings and abbreviations are used for the same journals (Angle Orthodontist and AJODO), please check also for upper and lower cases in the journal titles and for references 20-23 the author´s first names are not abbreviated like in the other references. Please check all references for consistency and copy&paste errors.
Author Response
Dear Reviewer,
Thank you for your comments. We checked all suggestions and comments and made the corresponding corrections on the article. Table A1 is below the conclusion.
Sincerely,
Silvia Capenakas (On behalf of the authors)
Reviewer 4 Report
Summary of manuscript: This manuscript reports a randomized clinical trial of comparing dental and skeletal changes obtained from Damon and RME expansion. This report concludes that the comparison between two groups showed significantly greater transverse expansion of the first molars and first premolars with buccal tipping in the RME group.
Specific comment
- It is natural that there is more skeletal expansion in the use of RME than with Damon. It is clear that there is greater transverse expansion of the first molars and first premolars in the RME because the Hylax appliance was attached to these teeth. This article is not novel as an original article.
Author Response
Dear Reviewer,
Thank you for your comments.
Sincerely,
Silvia Capenakas (On behalf of the authors)